# LEARNABLE COUNTERFACTUAL ATTENTION FOR SINGER IDENTIFICATION

## ABSTRACT

Counterfactual attention learning (Rao et al., 2021) utilizes counterfactual causality to guide attention learning and has demonstrated great potential in fine-grained recognition tasks. Despite its excellent performance, existing counterfactual attention is not learned directly from the network itself; instead, it relies on employing random attentions. To address the limitation, we target at singer identification (SID) task and present a learnable counterfactual attention (LCA) mechanism, to enhance the ability of counterfactual attention to help identify fine-grained vocals. Specifically, our LCA mechanism is implemented by introducing a counterfactual attention branch into the original attention-based deep-net model. Guided by multiple well-designed loss functions, the model pushes the counterfactual attention branch to uncover attention regions that are meaningful yet not overly discriminative (seemingly accurate but ultimately misleading), while guiding the main branch to deviate from those regions, thereby focusing attention on discriminative regions to learn singer-specific features in fine-grained vocals. Evaluation on the benchmark *artist20* dataset (Ellis, 2007) demonstrates that our LCA mechanism brings a comprehensive performance improvement for the deep-net model of SID. Moreover, since the LCA mechanism is only used during training, it doesn't impact testing efficiency.

## 1 INTRODUCTION

The human capacity to identify singers with only a handful of cues is nothing short of remarkable. With just a few musical snippets, people can swiftly discern a singer's unique vocal characteristics. This ability showcases the power of our auditory system: even amidst a backdrop of instrumental music, or faced with subtle sound variations presented by different singers (fine-grained vocals), we are still able to attend to specific frequency bands, comprehend the correlations among various audio features, and extract distinctive identifying cues to identify the singer.

The functionalities provided by the human auditory system dovetail with a particular class of deep learning algorithms, known as the *attention mechanism*. This mechanism empowers deep-net models to selectively focus attention on relevant things while disregarding extraneous ones, thus extracting distinctive identifying cues. More specifically, it primarily enables deep-net models to concentrate attention on discriminative regions to mitigate the negative effects caused by subtle sound variations from different singers and cluttered sound backgrounds, thereby extracting discriminative features to enhance the performance of the singer identification (SID) task (Kuo et al., 2021).

Despite the widespread adoption of the attention mechanism, effectively controlling models to learn correct attention remains largely unexplored. Most current methods are based on weak supervision, relying solely on the final classification loss for guidance (Rao et al., 2021). They lack strong, additional signals to steer the training process. More specifically, these methods often employ only softmax cross-entropy loss to supervise the final prediction, neglecting the essential causality between prediction and attention. When data bias is present, such a learning method can misdirect the attention mechanism to focus on inappropriate regions (induced by bias), thus negatively affecting task performance (Geirhos et al., 2020). This problem is especially pronounced in the SID task due to the pervasive background music. In such a situation, the model frequently tends to focus on the regions (*e.g.*, frequency bands) of background music and learn specific features from them, such as distinctive instrument sounds, rather than from the vocals. For instance, as Adele's songs often include piano accompaniment, the attention model is likely to consider the piano's frequency bands as discriminative regions. This tendency can impair the model's generalization ability, resulting in

a decline in its performance on the SID task. Additionally, the inherent variability in human vocal characteristics and the subtle differences in vocal organs between singers may also make it difficult for the attention mechanism to learn to focus on the appropriate regions. This, in turn, may diminish the discriminative ability of the extracted features. Considering these factors, it is crucial to guide the attention mechanism in the model to learn to focus on appropriate regions, in order to mitigate the impact of data bias and sound variability.

Recently, Rao et al. (2021) introduced a novel strategy for attention learning, termed counterfactual attention learning (CAL). This strategy involves the introduction of random attention maps as a form of counterfactual intervention for original attention maps. The underlying concept is to measure the quality of attentions by comparing the effect of actual learned attentions (facts) to that of random attentions (counterfactuals) on the final classification score, thereby encouraging the attention model to learn to concentrate on regions that are more beneficial and discriminative, consequently improving fine-grained recognition performance. Despite its excellent performance, this strategy faces certain limitations in mitigating the impact of biased data or sound variation, and in enhancing the model's generalization ability, owing to its dependence on using random attentions. To address these limitations, it's crucial to train the counterfactual attention to concentrate on meaningful regions (*e.g.,* specific biases such as the piano accompaniment in Adele's songs when considering the SID) rather than depending on random attentions as an intervention.

Building on the points mentioned above, to effectively enhance the learning of attention models, we posit that an ideal counterfactual attention should have the following characteristics: Firstly, its focused regions should be meaningful, such as those concentrating on biases, and should exhibit discriminative power in the learned features; Secondly, the features it learns should not have greater discriminative power for the task than those learned by the main branch's attention; Finally, the regions it focuses should differ from those highlighted in the main branch's attention.

Given the aforementioned characteristics, for the SID task, this study proposes a learnable counterfactual attention (LCA) mechanism. It incorporates a learnable counterfactual attention branch into the attention-based deep-net model, along with multiple well-designed loss functions (objectives). These objectives are designed to prompt the counterfactual attention branch to uncover attention regions that are meaningful but not overly discriminative (seemingly accurate but ultimately misleading), while guiding the main branch to shift away from these regions and thus focus on more appropriate regions for a better grasp of discriminative features in both fine-grained vocals and background music. Figure 1 illustrates that by incorporating the proposed LCA mechanism into the state-of-the-art SID model, CRNN_FGNL (Kuo et al., 2021), the model effectively bypasses data biases to focus on more relevant regions, enhancing its ability to capture discriminative features. Specifically, from the *artist20* dataset (Ellis, 2007), we randomly selected five artists to perform this experiment and introduced white noise in the 3k to 5k Hz range to recordings exclusively from one artist, Aerosmith (see Figure 1 (a)). In simple terms, the model can easily identify the artist (Aerosmith) by merely focusing on the noise (bias). Subsequently, we trained the CRNN_FGNL both with and without the LCA mechanism to visualize the class activation map (CAM) of the test data based on Grad-CAM (Selvaraju et al., 2017), specifically for the artist with added noise. Compared to Figure 1 (b) without using the LCA mechanism, it's evident from Figure 1 (c) that after incorporating the LCA mechanism, CRNN_FGNL significantly reduced its attention to the noise (bias) range. Additionally, to confirm the effectiveness of our LCA mechanism in enhancing the model's ability to learn discriminative features, we introduced the same noise to the test data of four other artists as was added to Aerosmith's recordings. Next, we employed t-SNE (van der Maaten & Hinton, 2008) for visualization and compared the feature distributions of CRNN_FGNL for these five artists, both with and without the use of the LCA mechanism. Compared to Figure 1 (d) without the LCA mechanism, Figure 1 (e) shows that with the LCA mechanism, CRNN_FGNL more effectively clusters each artist's features in the embedding space and distinguishes different artists, significantly mitigating the impact of noise. These results demonstrate the capability of the proposed LCA mechanism.

To the best of our knowledge, our study is the first to introduce counterfactual attention learning in addressing the SID task. Extensive experimental results demonstrate: 1) Our LCA mechanism brings a comprehensive performance improvement for the state-of-the-art SID model (Kuo et al., 2021); 2) The LCA mechanism is efficient, as it's applied only during training, ensuring no added computational load during testing.

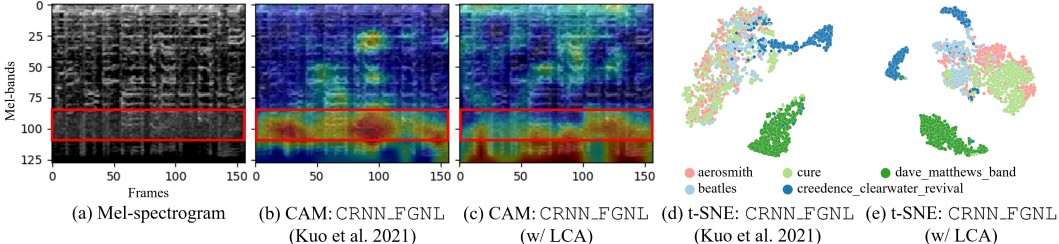

(a) Mel-spectrogram  (b) CAM: CRNN_FGNL  (c) CAM: CRNN_FGNL  (d) t-SNE: CRNN_FGNL  (e) t-SNE: CRNN_FGNL
                           (Kuo et al. 2021)        (w/ LCA)            (Kuo et al. 2021)        (w/ LCA)

Figure 1: (a) The 5-sec Mel-spectrogram with white noise (see red frame); (b) Class activation map (CAM) of CRNN_FGNL; (c) Class activation map (CAM) of our CRNN_FGNL (with LCA); (d) The t-SNE visualization of CRNN_FGNL; (e) The t-SNE visualization of our CRNN_FGNL (with LCA).

## 2 RELATED WORK

### 2.1 SINGER IDENTIFICATION

In music information retrieval, SID is a central task. Its primary objective is to identify the singer from a provided musical snippet or clip, thereby enhancing the systematic organization of musical archives. This task faces two primary challenges. Firstly, due to the inherent variability of human vocal characteristics and the subtle differences in vocal organs, a single singer can exhibit diverse singing tones. Conversely, multiple singers might possess strikingly similar voice qualities. This results in small inter-class variations and pronounced intra-class variations (Hsieh et al., 2020; Sundberg, 1987). As the number of singers considered increases, this issue intensifies. Secondly, since songs in a singer's albums typically include instrumental accompaniment, the SID model struggles to extract pure vocal features. The presence of background music can inadvertently be treated as an identification cue, compromising the model's ability to generalize across various recordings (Kuo et al., 2021; Hsieh et al., 2020; Van et al., 2019; Sharma et al., 2019; Rafii et al., 2018; Sturm, 2014).

With the advancements in deep learning, deep-net architectures such as convolutional neural networks, recurrent neural networks, and attention-based deep-net models have been widely used to tackle both challenges. For the first challenge, the core of these methods is to learn discriminative feature representations for the singers to be identified. For example, Nasrullah & Zhao (2019) introduced a convolutional recurrent neural network (CRNN) to better model the temporal correlations for the network to extract discriminative feature representations. Hsieh et al. (2020) extended the CRNN by integrating an additional branch dedicated to melody features, resulting in the CRNNM model with improved performance. Zhang et al. (2022) integrated timbre and perceptual features into the CRNN to improve its performance. Van et al. (2019) employed a bidirectional long short-term memory (LSTM) network to capture the temporal correlations within feature representations for SID. Kuo et al. (2021) integrated a new attention mechanism, the fully generalized non-local (FGNL) module, into the CRNN. This module captures correlations across positions, channels, and layers, thereby strengthening its capacity to learn non-local context relations (*i.e.,* long-range dependencies) in audio features, which in turn improves the identification of fine-grained vocals. Despite the great success of FGNL module used in SID, it is inherently trained with weak supervision, relying solely on the final classification (softmax cross-entropy) loss as guidance, neglecting the crucial causality between prediction and attention, limiting the potential for performance improvement.

For the second challenge, the key is to separate the vocal components from the given musical snippet or clip, thereby reducing the impact of instrumental sounds on the SID model. For example, Van et al. (2019) and Sharma et al. (2019) utilized U-Net architectures to learn the separation of singing voices, thereby reducing the impact of instrumental sounds on singer identity cues. Hsieh et al. (2020) and Kuo et al. (2021) employed Open-Unmix (Stöter et al., 2019), an open-source tool based on a three-layer bidirectional LSTM, to separate vocal and instrumental components in music. Building on the advancements in source separation technology and in line with previous works, we integrate the source separation model (Stöter et al., 2019) into our system. We aim to tackle the first challenge by introducing the counterfactual attention learning.

### 2.2 ATTENTION MECHANISM

The attention mechanism has gained popularity in sequence modeling due to its capacity to capture long-range dependencies and selectively focus on relevant parts of the input (Vaswani et al., 2017;

Bahdanau et al., 2015; Devlin et al., 2019; Yu et al., 2018; Guo et al., 2022). For example, Bahdanau et al. (2015) was the pioneer in introducing the attention mechanism into the sequence-to-sequence model for neural machine translation. This innovation enables the model to selectively attend to pertinent information from source sentences of varying lengths. Vaswani et al. (2017) unveiled the *Transformer* architecture that leverages a self-attention mechanism. Instead of relying on recurrent operations, this architecture employs a self-attention mechanism to analyze and update each element of a sentence using aggregated information from the whole sentence, leading to significant enhancements in machine translation outcomes. Its unprecedented success paved the way for a generation of Transformer-based models. Models based on this architecture or its variants have demonstrated exceptional results across multiple domains (Devlin et al., 2019; Radford et al., 2018; Wang et al., 2018; Huang et al., 2019; Ramachandran et al., 2019; Bello et al., 2019; Dosovitskiy et al., 2021; Kuo et al., 2021; Wei et al., 2022; Ng et al., 2023; Dong et al., 2023). Other attention mechanisms, such as channel-wise attentions (Hu et al., 2018; Woo et al., 2018; Roy et al., 2018), have also been developed to capture specific properties while training features. Despite the benefits of attention mechanisms, most existing methods learn it in a weakly-supervised manner, relying only on the final classification loss for supervision without considering other supervisory signals to guide the training process (Rao et al., 2021). This neglects the link (causality) between prediction and attention, easily causing data biases to mislead attention and harm task performance. Although Rao et al. (2021) recently introduced CAL that aims to bolster attention mechanism learning by using random attention maps as counterfactual interventions to the original attention maps, these random attention maps fail to accurately capture the inherent data biases, leading to limited performance gains. As a result, in this study, we focus on the SID task and introduce the LCA mechanism, which aims to learn meaningful counterfactual attentions as interventions for original attention maps to improve the performance of attention model.

## 3 APPROACH

### 3.1 REVISITING COUNTERFACTUAL ATTENTION LEARNING

We revisit the CAL in the form of the causal graph. A causal graph, also referred to as a structural causal model, is a directed acyclic graph with nodes $\mathcal{N}$ and causal links $\mathcal{E}$, denoted as $\mathcal{G} = \{\mathcal{N}, \mathcal{E}\}$. In Rao et al. (2021), the causal graph is formed with nodes that represent the variables in the attention model. These variables include the feature maps (or input data) $\boldsymbol{X}$, the learned attention maps $\boldsymbol{A}$, and the final prediction $\boldsymbol{y}$ as shown in Figure 2 (a). The arrow "$\rightarrow$" signifies the causal relations among these three nodes. For example, the link $\boldsymbol{X} \rightarrow \boldsymbol{A}$ indicates that the attention model takes feature maps as input and produces corresponding attention maps, while $(\boldsymbol{X}, \boldsymbol{A}) \rightarrow \boldsymbol{y}$ implies feature maps and attention maps jointly determine the final prediction. In CAL, random attention maps $\bar{\boldsymbol{A}}$ are introduced as counterfactual interventions for the original attention maps $\boldsymbol{A}$, resulting in the causal graph where $\boldsymbol{X} \rightarrow \boldsymbol{A}$ is replaced by $\boldsymbol{X} \dashrightarrow \bar{\boldsymbol{A}}$. The symbol "$\dashrightarrow$" represents cutting off the link between $\boldsymbol{X}$ and $\bar{\boldsymbol{A}}$, indicating that $\bar{\boldsymbol{A}}$ is not caused by $\boldsymbol{X}$. Then, $(\boldsymbol{X}, \bar{\boldsymbol{A}})$ jointly determines the final prediction $\bar{\boldsymbol{y}}$ based on counterfactual attentions. Based on the causal graphs, in CAL, the actual effect of the attention on the prediction, represented as $\boldsymbol{y}_{effect}$, is defined as the difference between the observed prediction $\boldsymbol{y}$ and its counterfactual $\bar{\boldsymbol{y}}$; specifically, $\boldsymbol{y}_{effect} = \boldsymbol{y} - \bar{\boldsymbol{y}}$. Finally, in addition to using the original standard classification loss, by further utilizing softmax cross-entropy loss $\mathcal{L}_{ce}$ to minimize the difference between $\boldsymbol{y}_{effect}$ and the ground truth classification label $\boldsymbol{y}_g$, the attention model is encouraged to focus attention on more discriminative regions, resulting in improved recognition accuracy (Rao et al., 2021).

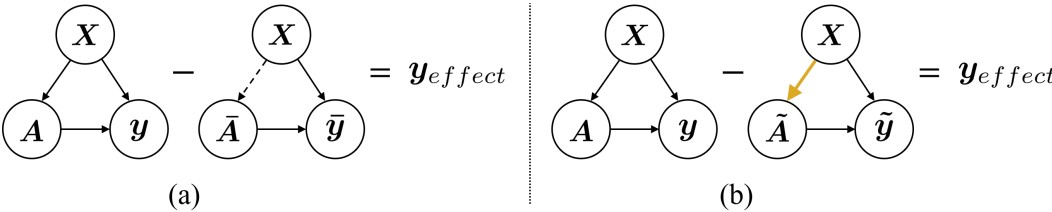

Figure 2: (a) Causal graph of CAL (Rao et al., 2021); (b) Causal graph of our LCA mechanism.

### 3.2 OUR LEARNABLE COUNTERFACTUAL ATTENTION MECHANISM

Consider that using random attention maps as counterfactual interventions may face certain limitations in mitigating the impact of biased data, as they lack the capability to identify meaningful (*e.g.,* biased) attention regions (seemingly accurate but ultimately misleading). Consequently, unlike Rao et al. (2021), in this study, we propose the LCA mechanism that integrates a learnable counterfactual attention branch into the attention model, specifically targeting biased attention regions that, while meaningful, can be deceptively discriminative (appearing accurate but ultimately misleading). In line of this, the causal graph representing the LCA mechanism is illustrated in Figure 2 (b), where $\tilde{\boldsymbol{A}}$ denotes the learned counterfactual attention maps that are causally related to $\boldsymbol{X}$; specifically, $\boldsymbol{X} \to \tilde{\boldsymbol{A}}$, and $(\boldsymbol{X}, \tilde{\boldsymbol{A}})$ jointly determine the final prediction $\tilde{\boldsymbol{y}}$. Similar to the CAL, in addition to using the original standard classification loss $\mathcal{L}_{ce}^{main} = \mathcal{L}_{ce}(\boldsymbol{y}, \boldsymbol{y}_g)$ in our main branch, by applying $\mathcal{L}_{ce}$ to minimize the difference between $\boldsymbol{y}_{effect}$ and $\boldsymbol{y}_g$ (the computed loss is referred to as $\mathcal{L}_{ce}^{effect}$), we penalize predictions driven by biased attentions. This forces the main branch of attention model to focus more on the primary cues instead of biased ones, thereby reducing the influence of a biased training set. Given this context, the pivotal issue is how to guide the counterfactual attention branch to focus on meaningful attention regions, such as those influenced by bias, which might appear accurate but could be misleading, in order to force the main branch in extracting specific (*e.g.,* singer-specific) discriminative features. To this end, we have, as previously mentioned, designed the following objective functions based on the three characteristics that ideal counterfactual attentions should possess.

**Characteristic 1: The focused regions of counterfactual attention should be meaningful.** To impart this characteristic, we use a standard classification loss to guide the learning of the counterfactual attention branch, defined as the softmax cross-entropy loss $\mathcal{L}_{ce}^{cf} = \mathcal{L}_{ce}(\tilde{\boldsymbol{y}}, \boldsymbol{y}_g)$. This loss ensures that the counterfactual attention branch has a certain level of classification capability, thereby encouraging the counterfactual attention to concentrate on meaningful regions. Compared to employing random attention maps for counterfactual intervention, the counterfactual attention maps learned from the counterfactual attention branch exhibit greater discriminative prowess. This, in turn, prompts the main branch to concentrate on regions with pronounced discriminative characteristics during training, particularly when concurrently minimizing loss $\mathcal{L}_{ce}^{effect}$. Specifically, under the guidance of $\mathcal{L}_{ce}^{effect}$, in order to minimize the difference between probability distributions $\boldsymbol{y}_{effect}$ and $\boldsymbol{y}_g$, given the improved prediction accuracy of $\tilde{\boldsymbol{y}}$ through the use of the learned counterfactual attention maps, a corresponding enhancement in $\boldsymbol{y}$'s prediction becomes essential. This is because $\boldsymbol{y}_{effect} = \boldsymbol{y} - \tilde{\boldsymbol{y}}$ implying that an enhancement in $\tilde{\boldsymbol{y}}$ necessitates a proportional improvement in $\boldsymbol{y}$ to satisfy the loss criteria. Additionally, by introducing the weight $\lambda_{ce}^{cf}$ to regulate the $\mathcal{L}_{ce}^{cf}$ in suppressing classification performance, we can reduce the tendency of counterfactual attention to overly focus on discriminative regions, ensuring its ability doesn't surpass the main branch.

**Characteristic 2: Targeting biased regions without outperforming the main branch.** Similar data biases frequently appear across multiple classes, rather than being confined to just one, resulting in multiple classes having substantial confidence scores (probabilities) when identifying a particular class. Take Adele's albums as an example: they often spotlight piano accompaniments. Yet, such piano traits are also present in the compositions of other artists. Consequently, when identifying Adele's song, extracting features from the piano frequency band may result in substantial confidence scores (probabilities) for multiple singers (classes) who also have songs accompanied by piano in the training set. Building upon this insight, to make the counterfactual attention branch to learn to focus on biased regions, we introduced an entropy loss denoted as $\mathcal{L}_{ent}$ for $\tilde{\boldsymbol{y}}$, resulting in $\mathcal{L}_{ent}^{cf} = \mathcal{L}_{ent}(\tilde{\boldsymbol{y}}) = -\sum_{c \in C} \tilde{y}_c \log \tilde{y}_c$, where $C$ is a set of all possible classes. By maximizing $\mathcal{L}_{ent}^{cf}$, the probability distribution of $\tilde{\boldsymbol{y}}$ is smoothed to be less peaked, directing the counterfactual attention more towards the biased regions. This in turn limits the ability of the counterfactual attention branch, ensuring it doesn't surpass the main branch.

**Characteristic 3: Regions of focus should differ from the main branch's attention.** To ensure that counterfactual attention and the main branch attention focus on distinct regions, we introduce a $\mathcal{L}_1$ loss to measure the difference between the counterfactual attention maps and the main branch attention maps as $\mathcal{L}_1^{att} = \mathcal{L}_1(\boldsymbol{A}, \tilde{\boldsymbol{A}}) = \frac{1}{N} \sum_{i=1}^{N} |A_i - \tilde{A}_i|$, where $N$ represents the total number of attention maps. By maximizing the $\mathcal{L}_1^{att}$, we ensure that the main branch and the counterfactual attention branch have distinct attention regions, thereby directing the main branch's attention away from biased regions.

**Total loss of LCA mechanism.** Beyond the aforementioned loss functions, we found that by integrating entropy loss with controlled weights (specifically smaller values) into the attention model's main branch, we can smooth out the probability distribution of classification during training, effectively alleviating the overfitting issue. As a result, we added the entropy loss to the main branch, represented as $\mathcal{L}_{ent}^{main} = \mathcal{L}_{ent}(\boldsymbol{y}) = -\sum_{c \in C} y_c \log y_c$ and controlled by $\lambda_{ent}^{main}$. Consequently, the total loss of the proposed LCA mechanism is represented as

$$\mathcal{L}_{total} = \lambda_{ce}^{main}\mathcal{L}_{ce}^{main} + \lambda_{ce}^{effect}\mathcal{L}_{ce}^{effect} + \lambda_{ce}^{cf}\mathcal{L}_{ce}^{cf} - \lambda_{ent}^{cf}\mathcal{L}_{ent}^{cf} - \lambda_{1}^{att}\mathcal{L}_{1}^{att} - \lambda_{ent}^{main}\mathcal{L}_{ent}^{main}, \quad (1)$$

where $\lambda_{ce}^{main}, \lambda_{ce}^{effect}, \lambda_{ce}^{cf}, \lambda_{ent}^{cf}, \lambda_{1}^{att}, \lambda_{ent}^{main}$ each represent the weight corresponding to their respective losses. By minimizing the total loss, the attention model zeroes in on regions with significant discriminative power, capturing unique features specific to the correct classes and simultaneously diminishing the impact of biased training data.

## 4 EXPERIMENTS

To demonstrate the effectiveness of the proposed LCA mechanism, we conduct SID experiments on the benchmark *artist20* dataset (Ellis, 2007). This dataset comprises $1,413$ complete songs spanning 20 different artists (singers). In the experiments, we utilize an album-split approach to ensure songs from the same album are exclusively allocated to the training, validation, or test set, to eliminate additional clues provided by the album (Kuo et al., 2021; Hsieh et al., 2020; Nasrullah & Zhao, 2019). All evaluated deep-net models are trained using audio snippets with lengths of 3s, 5s, and 10s, respectively (Kuo et al., 2021). Of these, 90% of the audio snippets (referred to as frames) are allocated for training, while the remaining 10% serve as the test set. The validation set is derived from 10% of the training set.

### 4.1 EVALUATION PROTOCOLS

We incorporate the proposed LCA mechanism into the state-of-the-art SID model, `CRNN_FGNL` (Kuo et al., 2021), and name it `CRNN_FGNL` (with LCA) for performance comparison. For `CRNN_FGNL`, we follow its original architecture settings as benchmark. Briefly, the `CRNN_FGNL` architecture is defined as a stack of four convolutional layers, one FGNL module, two gated recurrent unit (GRU) layers, and one fully connected (FC) layer. Within this architecture, the FGNL module (Kuo et al., 2021) itself consists of three sub-modules: a Gaussian filter, an non-local (NL) operation with rolling, and a modified squeeze-and-excitation (MoSE) scheme. For our `CRNN_FGNL` (with LCA), as shown in Figure 3, we insert an additional MoSE sub-module as a counterfactual attention branch within the FGNL module. For training, we use random initialization for the weights, set a constant learning rate of $10^{-4}$, employ dropout to prevent over-fitting, and utilize the Adam solver (Kingma & Ba, 2015) for optimization. The batch size is set to 16. Additionally, to ensure that the classification ability of the counterfactual attention branch does not exceed that of the main branch, we adopt a two-step training strategy: training the main branch for 20 epochs before initiating the training of the counterfactual attention branch. The entire network is trained for 800 epochs using an NVIDIA Tesla P40 GPU. Other hyperparameters, including the weight of each loss function, are determined based on the performance on the validation set. Note that our LCA mechanism introduces no additional computation during the testing process. Specifically, the counterfactual attention branch is removed during the testing phase. We will release the codes for more details.

To evaluate the impact of background accompaniment on the generalization ability of all compared deep-net models, we consider two evaluation settings: including the *original audio file* and the *vocal-only*. The distinction between the two is that, in the vocal-only setting, the Open-Unmix toolkit (Stöter et al., 2019) is used to separate the vocal components from each audio file during both training and testing. In the experiments, we report the evaluation results for each compared deep-net model including CRNN (Nasrullah & Zhao, 2019), CRNNM (Hsieh et al., 2020), `CRNN_FGNL` (Kuo et al., 2021), and our `CRNN_FGNL` (with LCA), at both the frame and song levels. Specifically, for frame-level evaluations, we treat each audio spectrogram of $t$-length (3s, 5s, or 10s) as an individual sample. Performance is then evaluated based on the F1 score calculated from all samples in the test set. For song-level evaluation, we employ a majority voting strategy (Kuo et al., 2021; Nasrullah

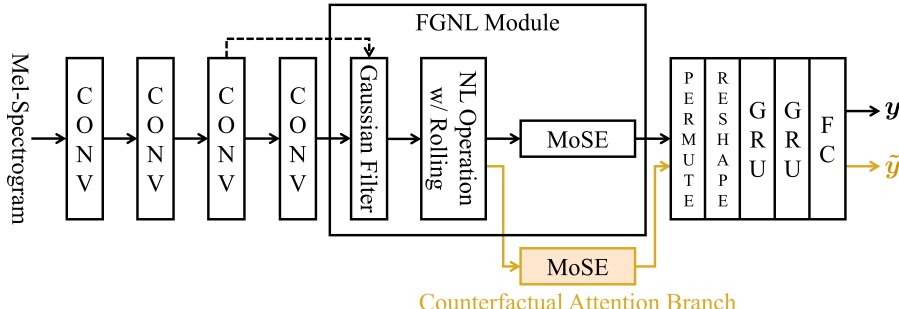

Figure 3: Proposed `CRNN_FGNL` (with LCA) architecture. The orange indicates the counterfactual attention branch, present only during the LCA mechanism's training phase and removed for testing.

Table 1: Ablation study of `CRNN_FGNL` employing CAL and our LCA mechanism with varied loss settings (utilizing 5-sec original audio files). Values in bold indicate the best performance.

| Models | Frame Level | | Song Level | |
|---|---|---|---|---|
| | Avg. | Best | Avg. | Best |
| `CRNN_FGNL`($\mathcal{L}_{ce}^{main}$) (Kuo et al., 2021) | 0.53 | 0.55 | 0.70 | 0.77 |
| `CRNN_FGNL` (w/ CAL) ($\mathcal{L}_{ce}^{main} + \mathcal{L}_{ce}^{effect}$) (Rao et al., 2021) | 0.53 | 0.55 | 0.70 | 0.77 |
| `CRNN_FGNL` (w/ LCA) Ours ($\mathcal{L}_{ce}^{main} + \mathcal{L}_{ce}^{effect} + \mathcal{L}_{ce}^{cf}$) | **0.56** | 0.56 | 0.73 | 0.80 |
| `CRNN_FGNL` (w/ LCA) Ours ($\mathcal{L}_{ce}^{main} + \mathcal{L}_{ce}^{effect} + \mathcal{L}_{ce}^{cf} - \mathcal{L}_{ent}^{cf} - \mathcal{L}_{ent}^{main}$) | **0.56** | **0.58** | 0.73 | 0.80 |
| `CRNN_FGNL` (w/ LCA) Ours ($\mathcal{L}_{ce}^{main} + \mathcal{L}_{ce}^{effect} + \mathcal{L}_{ce}^{cf} - \mathcal{L}_{ent}^{cf} - \mathcal{L}_{ent}^{main} - \mathcal{L}_1^{att}$) | 0.55 | 0.57 | **0.74** | **0.81** |

& Zhao, 2019), where the most frequently predicted artist at the frame level is selected as the final prediction for each song. Performance is then quantified at the song level by reporting the F1 score.

## 4.2 ABLATION ANALYSIS

Before comparing our method with existing state-of-the-art SID methods, we first conduct ablation experiments to validate the effectiveness of the loss terms in the proposed LCA mechanism. Specifically, we first adopt `CRNN_FGNL` as the baseline for comparison and incorporate the CAL (Rao et al., 2021). This combined model is referred to as `CRNN_FGNL` (with CAL). That is, we employ random attention maps as counterfactual interventions to the original attention maps in MoSE sub-module to evaluate the effect of CAL on the performance of `CRNN_FGNL` in SID. Next, we incorporate our LCA mechanism into the `CRNN_FGNL` architecture (see Figure 3) and introduce the $\mathcal{L}_{ce}^{cf}$ as described in **Characteristic 1** (see sub-section 3.2). Through this, we aim to learn counterfactual attentions and evaluate their effect on the performance of `CRNN_FGNL` in the SID task. We subsequently incorporate the loss terms $\mathcal{L}_{ent}^{cf}$, $\mathcal{L}_1^{att}$, as outlined in **Characteristics 2** and **3**, as well as $\mathcal{L}_{ent}^{main}$, to further evaluate their contributions on performance. All ablation experiments were conducted on the original audio file setting with 5s length, and considered the average and best test F1 scores at frame level and song level from three independent runs. Note that to enhance the efficiency of ablation experiments, we adopted an early stopping strategy during training. For the sake of concise representation in Table 1, we omitted the weight notation before each loss term.

The results in Table 1 first demonstrate that incorporating CAL (*i.e.,* `CRNN_FGNL` (with CAL)) does not improve performance compared to `CRNN_FGNL`. The reason may be that singing voices are distributed in most frequency bands in the spectrogram, while in images, target objects usually only occupy a small portion of the frame, most of which is irrelevant background. Therefore, in contrast to CAL's exceptional performance in image-related tasks (Rao et al., 2021), the effectiveness of using random attention maps to target positions on the spectrogram as counterfactual interventions for the SID task is relatively limited. Compared to using random attention maps for counterfactual intervention, after introducing discriminative counterfactual attention maps through our LCA mechanism (*i.e.,* with $\mathcal{L}_{ce}^{cf}$), the main branch is compelled to focus attention on more discriminative regions in order to satisfy the loss $\mathcal{L}_{ce}^{effect}$, thereby enhancing the performance of SID. By further

Table 2: Ablation study of `CRNN_FGNL` (with LCA) with and without two-step training strategy.

| | Frame Level | | Song Level | |
|---|---|---|---|---|
| Models | Avg. | Best | Avg. | Best |
| `CRNN_FGNL` (w/ LCA) - w/o two step training | **0.56** | **0.58** | 0.72 | 0.77 |
| `CRNN_FGNL` (w/ LCA) - w/ two step training | 0.55 | 0.57 | **0.74** | **0.81** |

Table 3: Quantitative evaluation of state-of-the-art SID models at both frame and song levels across different length settings (3s, 5s, or 10s), considering both original audio file and vocal-only settings.

| | | Original Audio File | | | | | | Vocal-only | | | | | | |
|---|---|---|---|---|---|---|---|---|---|---|---|---|---|---|
| | | Frame Level | | | Song Level | | | Frame Level | | | Song Level | | | #Para. |
| Models | | 3s | 5s | 10s | 3s | 5s | 10s | 3s | 5s | 10s | 3s | 5s | 10s | (M) |
| CRNN | Avg. | 0.44 | 0.45 | 0.48 | 0.57 | 0.55 | 0.58 | 0.42 | 0.46 | 0.51 | 0.72 | 0.74 | 0.74 | 0.39 |
| (Nasrullah & Zhao, 2019) | Best | 0.46 | 0.47 | 0.53 | 0.62 | 0.59 | 0.60 | 0.44 | 0.48 | 0.53 | 0.76 | 0.79 | 0.77 | |
| CRNNM | Avg. | 0.47 | 0.47 | 0.51 | 0.62 | 0.61 | 0.65 | 0.42 | 0.46 | 0.49 | 0.73 | 0.75 | 0.73 | 0.78 |
| (Hsieh et al., 2020) | Best | 0.48 | 0.50 | 0.53 | 0.67 | 0.68 | 0.69 | 0.43 | 0.47 | 0.50 | 0.75 | 0.79 | 0.75 | |
| CRNN_FGNL | Avg. | 0.52 | 0.54 | 0.55 | 0.72 | 0.73 | 0.73 | **0.44** | 0.47 | 0.51 | 0.79 | 0.80 | 0.79 | 0.58 |
| (Kuo et al., 2021) | Best | 0.54 | 0.57 | 0.58 | 0.76 | 0.79 | 0.78 | 0.44 | 0.48 | 0.53 | 0.81 | 0.82 | **0.83** | |
| CRNN_FGNL (w/ LCA) | Avg. | **0.53** | **0.56** | **0.59** | **0.74** | **0.78** | **0.75** | **0.44** | **0.51** | **0.54** | **0.80** | **0.81** | **0.80** | 0.58 |
| Ours | Best | **0.56** | **0.59** | **0.62** | **0.79** | **0.80** | **0.80** | **0.45** | **0.51** | **0.56** | **0.83** | **0.83** | **0.83** | |

introducing entropy loss $\mathcal{L}_{ent}^{cf}$ to direct the counterfactual attention branch to focus on biased regions, reducing overfitting in the main branch with $\mathcal{L}_{ent}^{main}$, and incorporating $\mathcal{L}_1^{att}$ to ensure distinct focus regions for both counterfactual and main branch attentions, we observe performance gains at both the frame-level and song-level. These results validate the effectiveness of the loss terms design in our LCA mechanism. Additionally, to verify whether introducing the two-step training strategy to suppress the performance of the counterfactual attention branch in the LCA mechanism can positively influence the main branch, we conduct ablation experiments on the `CRNN_FGNL` (with LCA), comparing models trained with and without this strategy. The results from Table 2 indicate that the overall performance of using the two-step training strategy surpasses that of not using it (*i.e.,* training the main branch and counterfactual attention branch simultaneously), especially in song-level performance. Therefore, based on the insights from the ablation study, in our subsequent experiments, we will fully train our model (`CRNN_FGNL` (with LCA)) without employing the early stopping strategy. The model will incorporate all the aforementioned loss terms and utilize the two-step training strategy, and then compare its performance with the current state-of-the-art SID models.

### 4.3 COMPARISON WITH STATE-OF-THE-ART SID MODELS

In this experiment, similar to the ablation study, Table 3 summarizes the average and best test F1 scores at both the frame and song levels for all comparison models, based on three independent runs. As our experimental setup is in line with `CRNN_FGNL`, for fairness, values of compared models are sourced directly from the reports of `CRNN_FGNL` (Kuo et al., 2021). In comparing CRNN and CRNNM, as shown in Table 3, our findings align with those of Hsieh et al. (2020), demonstrating that CRNNM outperforms CRNN in most settings, especially in the original audio file setting. This confirms that incorporating melody features can indeed enhance the performance of the SID task on original audio files with instrumental accompaniment. However, when the instrumental accompaniment is removed using the source separation technique (Stöter et al., 2019), its effectiveness becomes limited and may even produce adverse effects (see the results of vocal-only setting). On the other hand, to incorporate melody features, the parameter count of CRNNM significantly increases. In addition, since both CRNN and CRNNM utilize only convolutional and recurrent operations without employing the self-attention mechanism, they face challenges in capturing non-local context relations (*i.e.,* long-range dependencies) (Vaswani et al., 2017; Wang et al., 2018) between audio features. This limitation also hampers their overall performance. Considering the aforementioned

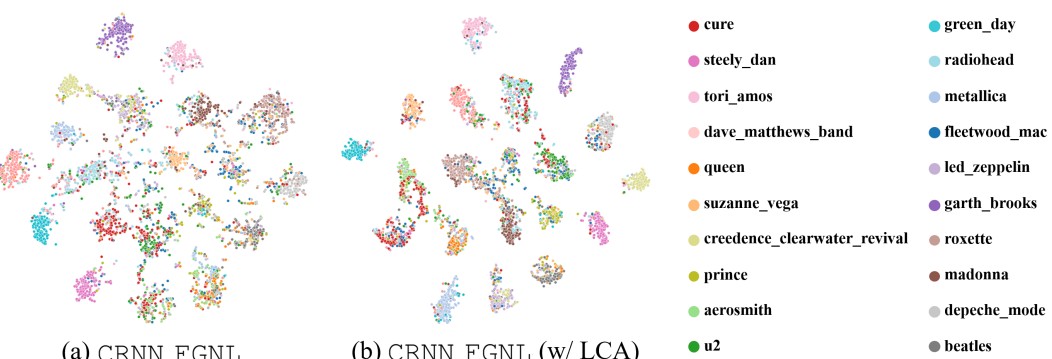

(a) CRNN_FGNL          (b) CRNN_FGNL (w/ LCA)

Figure 4: The t-SNE visualization of the features for (a) CRNN_FGNL (Kuo et al., 2021) and (b) CRNN_FGNL (with LCA) under the original audio file setting of the 10-sec frame-level test samples.

Table 4: Quantitative evaluation of CRNN_FGNL (with LCA)'s main branch and counterfactual attention branch. Evaluation settings are identical to the original audio file setting in Table 3.

| | | **Frame Level** | | | **Song Level** | | |
|---|---|---|---|---|---|---|---|
| CRNN_FGNL (w/ LCA) | | 3s | 5s | 10s | 3s | 5s | 10s |
| Counterfactual Attention Branch | Avg. | 0.20 | 0.16 | 0.41 | 0.26 | 0.21 | 0.51 |
| Main Branch | Avg. | **0.53** | **0.56** | **0.59** | **0.74** | **0.78** | **0.75** |

issues, Kuo et al. (2021) introduced the self-attention mechanism and proposed the FGNL module. This module is designed to further model the correlations among all of the positions in the feature map across channels and layers. Even without the incorporation of melody features, integrating the FGNL module with CRNN (denoted as CRNN_FGNL) significantly enhances the performance of SID. Although CRNN_FGNL achieves outstanding performance in SID, it relies solely on the standard classification loss for supervision without considering other supervisory signals to guide the training process. This overlooks the causality between prediction and attention. To tackle the issue, we proposed the LCA mechanism to explicitly link the causality between prediction and attention. By further incorporating the LCA mechanism, results support that CRNN_FGNL (with LCA) is able to concentrate attention on more discriminative regions, enabling it to learn singer-specific discriminative features for SID. The t-SNE (van der Maaten & Hinton, 2008) visualization of the 10-sec frame-level original audio files further confirms that the features extracted from our CRNN_FGNL (with LCA) can distinguish between different singers more effectively, in contrast to the CRNN_FGNL which appears more divergent and less discriminative (see Figure 4). Compared to the original CRNN_FGNL, as shown in Table 3, CRNN_FGNL (with LCA) achieves comprehensive performance improvements, underscoring its superior generalization ability. Additionally, the LCA mechanism is used only during training. As a result, it does not affect the final parameter count of the model, meaning that CRNN_FGNL (with LCA) and CRNN_FGNL have the same number of parameters (see Table 3). Finally, we evaluate the performance of the counterfactual attention branch to confirm whether it truly does not exceed the performance of the main branch. The results in Table 4 demonstrate that, taking into account the two-step training, characteristics of the loss function design (*e.g.,* entropy loss $\mathcal{L}_{ent}^{cf}$), and the loss weights, the performance of the proposed counterfactual attention branch is constrained and indeed does not exceed that of the main branch.

## 5 CONCLUSION

We introduced a learnable counterfactual attention (LCA) mechanism, a new counterfactual attention learning approach, to guide the attention model in focusing attention on more discriminative regions to learn the singer-specific features to help identify fine-grained vocals. Extensive experimental results on the benchmark dataset, artist20 (Ellis, 2007) indicate that the proposed LCA mechanism significantly improves the accuracy of the deep-net model in singer identification (SID) task and achieves the state-of-the-art level. Building upon these encouraging results, our upcoming efforts will be dedicated to the development of learnable data augmentation techniques to improve the fineness of the learned feature representation and its generalization ability.

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
