# OpenReview forum: "Learnable Counterfactual Attention for Singer Identification"
_ICLR.cc/2024/Conference — Submitted to ICLR 2024_

### Official Review · Reviewer_6GjB · 2023-10-30

**Soundness:** 3 good
**Presentation:** 3 good
**Contribution:** 3 good
**Rating:** 6
**Confidence:** 1

**Summary:**

This paper is out of my knowledge, and I tend to not submit any reviews for this paper. Thanks for the submission. Please ignore my ratings.

**Strengths:**

This paper is out of my knowledge, and I tend to not submit any reviews for this paper. Thanks for the submission.

**Weaknesses:**

This paper is out of my knowledge, and I tend to not submit any reviews for this paper. Thanks for the submission.

**Questions:**

This paper is out of my knowledge, and I tend to not submit any reviews for this paper. Thanks for the submission.

---

> ### Author Response · Authors · 2023-11-19
> **Response to the Reviewer 6GjB**
>
> Thank you.

---

### Official Review · Reviewer_i4Hf · 2023-10-30

**Soundness:** 2 fair
**Presentation:** 2 fair
**Contribution:** 2 fair
**Rating:** 3
**Confidence:** 5

**Summary:**

This paper proposes to extend counterfactual attention learning via a learnable counterfactual attention module to further improve the ability of counterfactual attention. To learn this learnable attention model, this paper designs a series of loss functions. Beyond the conventional cross-entropy and counterfactual losses, it adds three extra loss terms. First, a cross-entropy loss is applied to the counterfactual attention branch as the regularization term to make the counterfactual attention more meaningful. Second, a multiple classification loss for the counterfactual attention branch is designed to limit the performance of the counterfactual attention. Third, an L1 loss between the attention maps of factual and counterfactual branches encourages the difference.  This method is evaluated with the singer identification task which also requires the fine-grained identification ability.   Specifically, the benchmark artist20 dataset is employed for the comparison, including the comparison with other SOTA methods, and ablation studies.

**Strengths:**

1) This paper proposes a learnable counterfactual attention module to achieve better performance.
2) This paper provides detailed explanations of each extra loss-term.

**Weaknesses:**

Some concerns about this paper are summarized below:
1) From the experiments in the original CAL method. It seems that the refinement of the counterfactual attention branch didn't improve the recognition performance. What is the motivation to refine the counterfactual attention branch? Did you find any insights from the preliminary experiments to support this enhancement path?
2) As shown in Figure 2 and Section 3.2, CAL applies the counterfactual "intervention" to cut off the causal relations between image and attention.  The goal of this "intervention" is to obtain the independent effects and further benefit the calculation of the Total Direct Effect. Here, both the counterfactual and factual branches are learned from the image. From the perspective of causality, how can you guarantee the independence between counterfactual and factual attention for the causal inference?
3) For the extra loss terms, there are two contradictory losses, one is to make the counterfactual attention meaningful and the other is to make it not too meaningful.  How can we balance these two losses?  Is it robust for different scenarios? Do the hyper-parameters of loss rates matter for the performance?
4)  The motivation of this paper is to improve counterfactual attention learning. Given this goal, it is better to evaluate the proposed method and the key baseline (CAL) in the same settings, such as CUB, Cars, Aircraft, and so on.
5) As shown in Table 1, the improvement of the proposed LCA is very limited. Why these results can support the conclusion? Take the Best frame-level results as an example, the proposed method only improves 0.02 accuracy.
6) Only one dataset is used.  The evaluation process seems to be not solid.  It is suggested to add more.

**Questions:**

Beyond the questions in the weaknesses part, there are some questions for the details.

1) As shown in Table 1, which counterfactual attention is used as a baseline, random, mean, or shuffle?
2) What are the loss rates used in the experiments?

---

> ### Author Response · Authors · 2023-11-19
> **Response to the Reviewer i4Hf**
>
> For Weakness 1:
>
> Thank you for your comment. Yes, experiments (Table 1) show that CAL's random attention maps are ineffective for SID task. This is due to the distinct nature of music tasks compared to vision tasks. For instance, in the SID task, singing voices are distributed in most frequency bands in the spectrogram, whereas in the image classification task, the target object usually occupies only a small portion of the image. This leads to the ineffectiveness of using random attention maps as counterfactual interventions in CAL for SID task. We addressed this by developing the LCA mechanism, which uses learnable attention maps to target specific biases instead of CAL's random attention maps. This approach improves SID and music genre classification performance (see Table 3 in original manuscript and Table A1 in appendix).
>
>  For Weakness 2:
>
> Our study, as shown in Table 1, reveals that using random attention maps as counterfactual interventions to cut off the causal link between feature maps and attention is ineffective for the SID task. We believe that forcibly maintaining such independence in model training is not always advantageous. We suggest that a data-driven approach, where counterfactual attentions could automatically learn from data, identify biased regions, could be more advantageous. This approach was validated in SID experiments (Table1 and 3) and further supported by music genre classification experiments (Table A1 in the appendix), proving that our LCA mechanism's strategy of linking feature maps and attention (i.e., dependence assumption) outperforms the CAL mechanism in music tasks.
>
> For Weakness 3:
>
> Thank you. We balance the two loss functions in our counterfactual attention branch by adjusting the hyperparameters $\lambda_{ce}^{cf}$ and $\lambda_{ent}^{cf}$ (see formula 1). In fact, in the SID task, we did not extensively experiment with these hyperparameters, setting them based on the original audio files and vocal-only tracks, without considering duration (3, 5, or 10 seconds) (see Table 3). For the new music genre classification task (see Table A1 in appendix), we also employed the same hyperparameters used in the SID task for original audio files, and achieved excellent performance, demonstrating our method's robustness. We think designing ablation experiments for various tasks to optimize hyperparameters for loss functions can enhance performance. However, from our experience, satisfactory results are achievable in our design as long as $\lambda_{ce}^{cf}$ is greater than $\lambda_{ent}^{cf}$.
>
> For Weakness 4:
>
> This study focuses on evaluating the CAL mechanism's efficacy in music-related tasks. In our experiments (see Table 1), we have demonstrated the LCA mechanism's superior efficacy over CAL in the SID task (a fine-grained challenge (Kuo et al., 2021)) under identical experimental conditions. Additionally, we confirmed our LCA mechanism's superiority over CAL in music genre classification (see appendix).
>
> For Weakness 5:
>
> Thank you for your comment. Table 1 shows our LCA mechanism comprehensively outperforms both the original CRNN\_FGNL and CRNN\_FGNL (with CAL) at frame and song levels. Table 3 further indicates that LCA enhances CRNN\_FGNL's performance in various settings, including 3, 5, and 10-second experiments with original and vocal-only audio. Additionally, the new music genre classification experiments in the appendix (see Table A1) reveal marked improvements with our LCA compared to CAL. Notably, our LCA is used only during training, thus not increasing the model's parameters or computational load in testing phase.
>
> For Weakness 6:
>
> We have included a music genre classification task in the appendix, which serves to further assess the performance of our method.
>
> For Question 1:
>
> In our original manuscript, we have clearly described in the results section (section 4.2) for Table 1 that we adopted random attention in the CAL mechanism. This approach of using random attention is also recommended by the original authors of the CAL mechanism (Rao et al., 2021).
>
> For Question 2:
>
> In the SID task, the weight of each loss function is as follows:
>
> Original audio file setting:
> $\lambda_{ce}^{main}$=  1.0,
> $\lambda_{ce}^{effect}$= 1.0,
> $\lambda_{ce}^{cf}$= 0.3,
> $\lambda_{ent}^{cf}$= 0.25,
> $\lambda_{1}^{att}$= 1.0,
> $\lambda_{ent}^{main}$= 0.2
>
> Vocal only setting:
> $\lambda_{ce}^{main}$=  1.0,
> $\lambda_{ce}^{effect}$= 1.0,
> $\lambda_{ce}^{cf}$= 0.8,
> $\lambda_{ent}^{cf}$= 0.025,
> $\lambda_{1}^{att}$= 1.0,
> $\lambda_{ent}^{main}$= 0.02
>
> For the newly implemented music genre classification task, we have set the weight of each loss function to be the same as those used in the original audio file setting of the SID task. We have added the hyperparameter setting (weight) of each loss function in the appendix. Thank you.

---

> > ### Comment · Reviewer_i4Hf · 2023-11-22
> > **Thanks for your response.**
> >
> > Thank you for your response, and I apologize for my own delay in replying.
> >
> > After carefully reviewing the current revision, I still have significant concerns that need to be addressed:
> >
> > Experiments and Comparisons:
> > The paper lacks comparative experimental data between CAL and LCA in the context of fine-grained recognition tasks. Moreover, the reasons or motivations supporting this choice remain unconvincing. Given the introduction of more hyperparameters, the improvements demonstrated on SID tasks are not persuasive enough. A more rigorous experimental comparison and justification are needed to validate these claims.
> >
> > Soundness and Causal Inference:
> > The implementation of the study does not seem to align with standard causal inference methodologies. It's crucial to provide new guarantees or robust evidence regarding the learned counterfactual attentions. Currently, the limited scope of the experiments conducted does not suffice to convincingly demonstrate the effectiveness of your approach. A more comprehensive and theoretically sound exploration is required to substantiate your findings.
> >
> > I hope these points will be thoroughly considered and addressed in your next revision. I believe they will help the quality of this paper.

---

> > > ### Author Response · Authors · 2023-11-23
> > > **Response to the Reviewer i4Hf**
> > >
> > > Thank you for your response. We have demonstrated the effectiveness of our method in the fine-grained SID task. Moreover, to further validate our LCA mechanism, we have added an experiment on music genre classification in the supplementary materials (with hyperparameter settings identical to the SID task), which also confirms the efficacy of the LCA mechanism. In both experiments, we provide not only data comparisons but also t-SNE visualizations of the features!

---

### Official Review · Reviewer_5eKQ · 2023-10-31

**Soundness:** 3 good
**Presentation:** 3 good
**Contribution:** 2 fair
**Rating:** 3
**Confidence:** 4

**Summary:**

This paper presents a learnable counterfactual attention method to improve the recognition performance of the singer identification task. The method improves the existing counterfactual attention learning framework by replacing the random counterfactual attention with learnable ones. With some new and specifically designed loss functions, the method can better discover effective attention regions and show improvement on SID benchmarks.

**Strengths:**

- The proposed learnable counterfactual attention is well-motivated and reasonable. Empirical results show the method is effective on the SID task.

- The paper is well-written and easy to follow.

**Weaknesses:**

- My main concern is the generality of the proposed method. The proposed method is motivated by the limitations of the existing CAL method. However, this paper only evaluates the method on the SID task, which is a less popular and competitive task compared to the tasks considered in CAL. According to the analysis provided in Section 3.2, LCA didn't add assumptions on the data or task types over CAL. So it is not clear why the proposed method is only evaluated on the SID task. Considering ICLR is a machine learning conference, I think showing the generality of the proposed method is also helpful to make the paper more suitable for publishing on ICLR and interest the audience of the conference.

- The authors claim the method can "guide the main branch to deviate from those regions, thereby focusing attention on discriminative
regions to learn singer-specific features in fine-grained vocals". I think it would be better to provide some quantitative evidence to support this claim.

**Questions:**

Although I find the proposed method is well-motivated and reasonable, I still have some concerns about the experimental study and positioning of this paper. I think the paper can be further improved if the above problems can be solved.

---

> ### Author Response · Authors · 2023-11-19
> **Response to the Reviewer 5eKQ**
>
> For Weakness 1:
>
> Thank you for your comment. Our study primarily investigates the efficacy of the CAL mechanism in music-related tasks, particularly in the SID task. Through experiments (as shown in Table 1), we found that the CAL mechanism is ineffective for the SID task. This is due to the distinct nature of music-related tasks compared to vision tasks. For instance, in the SID task, singing voices are distributed in most frequency bands in the spectrogram, whereas in the image classification task, the target object usually occupies only a small portion of the image. This leads to the ineffectiveness of using random attention maps as counterfactual interventions in CAL for SID task. To overcome this limitation, we developed the LCA mechanism. It replaces the random attention maps used as counterfactual interventions in CAL with attention maps that automatically focus on biased regions, generated by a learnable counterfactual attention branch. This improvement has been proven effective in enhancing the SID task in experiments (see Table 3).
>
> On the other hand, as you pointed out, the LCA mechanism does not add assumptions about data or task types compared to CAL. This is because our initial design intention of the LCA mechanism was to allow it to learn from data, thereby generating counterfactual attention maps that benefit the model. We expect this design to be more easily applicable to different tasks, thereby enhancing its universality.
>
> Finally, the SID task is a core topic in the field of Music Information Retrieval (MIR). However, following your valuable suggestion, we have added a task for music genre classification in the appendix to validate the universality and effectiveness of our proposed LCA mechanism in music-related tasks.
>
> For Weakness 2:
>
> Thank you for your comment. In Table 3 of our original manuscript, we have already demonstrated that integrating the LCA mechanism into the CRNN\_FGNL model (i.e., CRNN\_FGNL (w/ LCA)) leads to a comprehensive enhancement in performance. This is attributed to the counterfactual attention branch learned from the LCA mechanism, which effectively redirects the main branch from less discriminative regions (bias regions) to those with higher discriminative potential (singer-specific regions). Moreover, the t-SNE visualization in Figure 4 further bolsters our assertion that CRNN\_FGNL (w/ LCA) demonstrates a significantly stronger discriminative ability in the learned features (i.e., can distinguish between different singers more effectively), as compared to the original CRNN\_FGNL model.
>
> Additionally, in the appendix, we have included an experiment to further confirm that the counterfactual branch can focus on common data biases (see Figure A1), thus encouraging the main branch to deviate from these biases and enhancing the discriminative nature of its features.
>
> For Question:
>
> Thank you for your valuable suggestion. Following your advice, we have added two additional experiments in the appendix to further validate the universality of our LCA mechanism and substantiate its claims.

---

### Official Review · Reviewer_hb43 · 2023-10-31

**Soundness:** 2 fair
**Presentation:** 3 good
**Contribution:** 2 fair
**Rating:** 6
**Confidence:** 3

**Summary:**

This paper introduces a learnable counterfactual attention mechanism, specifically tailored for the singer identification task, aiming to address the limitations of existing counterfactual attention learning. Unlike the traditional approach that depends on random attentions, the LCA mechanism enhances the ability of counterfactual attention to identify fine-grained vocals through direct learning. The implementation involves integrating a counterfactual attention branch into the existing model. This addition is meticulously guided by multiple loss functions, ensuring that the counterfactual attention branch focuses on regions that are meaningful yet not overly discriminative, avoiding potentially misleading results. Meanwhile, it directs the main branch towards discriminative regions to learn singer-specific features effectively. The performance improvement is demonstrated through evaluation on the benchmark artist20 dataset.

**Strengths:**

The authors provide a clear and intuitive analysis of the limitations present in the baseline method (CAL), proposing straightforward yet effective solutions to enhance its performance. Each proposed solution, addressing Characteristics 1, 2, and 3, demonstrates simplicity and efficacy. The experimental results solidify the effectiveness of the introduced loss functions, providing tangible evidence of improvement. Furthermore, the manuscript is well-structured and articulated, ensuring a smooth and comprehensible reading experience for the audience.

**Weaknesses:**

1. Regarding the second Characteristic 2 (Targeting biased regions without outperforming the main branch), I can not see the logic between forcing the class distribution to be smooth and targeting biased regions without outperforming the main branch. Moreover, the assertion that a smoother output distribution directly correlates to effectively targeting biased regions without overshadowing the main branch is questionable. Since the output distributions of the two branches lead to disparate final classification results, this assumption appears to be unfounded and requires further clarification.

2. Regarding Characteristic 3 (Regions of focus should differ from the main branch’s attention), the manuscript appears to implicitly assume superior performance of the main branch. However, empirical observations suggest that the main branch’s performance is not as exemplary as presumed. A straightforward deviation from the main branch's attention map might inadvertently introduce inaccuracies. This aspect of the methodology warrants a more cautious approach and a thorough examination to validate its effectiveness and mitigate potential risks of error introduction.

**Questions:**

Please refer to the weaknesses.

---

> ### Author Response · Authors · 2023-11-19
> **Response to the Reviewer hb43**
>
> For Weakness 1:
>
> Thank you for your comment. In the design of neural network training, if entropy is used as the loss function, then when the probability distribution of the output layer becomes more smooth or uniform— that is, the probabilities of each category tend to be consistent — the entropy value will reach its maximum. Under this strategy of pursuing maximum entropy in Characteristic 2, the neural network is driven to focus on the common biases (data biases) in the training data across all categories, as focusing on common data biases will more easily lead the model's output layer to produce smooth or uniform probabilities across different categories. This approach of concentrating on biased regions means that the counterfactual branch is unlikely to surpass the main branch in terms of classification performance, as these bias regions typically lack strong discriminative power. Consequently, by further incorporating the loss function outlined in Characteristic 3, which is designed to maximize the differences between the counterfactual and main branch attention maps, our approach guides the main branch to shift its focus away from biased regions and instead concentrate on regions with greater discriminative power.
>
> To validate that incorporating the loss functions of Characteristic 2 and 3 in the proposed learnable counterfactual attention (LCA) mechanism enables the counterfactual branch to focus on common data biases, thereby prompting the main branch to deviate from these biases, we expanded the experiment shown in Figure 1. Specifically, we introduced the same white noise in the 3k to 5k Hz range of the spectrogram for five different singers at varying proportions. Subsequently, we trained the CRNN\_FGNL both with and without the LCA mechanism, visualizing the class activation maps (CAM) of the test data based on Grad-CAM. As can be seen in Figure A1 of the appendix, compared to Figure A1 (b) without the LCA mechanism, Figure A1 (c) clearly shows that after integrating the LCA mechanism, CRNN\_FGNL significantly reduced its attention to the noise (bias) range. Such results confirm the effectiveness of our LCA mechanism in guiding the counterfactual attention branch to concentrate on common biased regions, subsequently prompting the main branch to divert its focus from these regions.
>
> For Weakness 2:
>
> Thank you for your comment. During our model training process, we intentionally designed the counterfactual attention branch to perform less effectively than the main branch. This is to ensure that the counterfactual attention branch focuses on learning the biases present in the data, specifically the regions with weaker discriminative power, thereby compelling the main branch to avoid concentrating on these biased regions. The experimental results presented in Table 4 of our manuscript confirm our design approach: the performance of the counterfactual branch is significantly lower than that of the main branch, indicating that the regions it focuses on have lower discriminative capabilities. Based on this, we further emphasize maximizing the differences in the attention maps between the counterfactual and main branches, aiming to direct the main branch's focus to regions distinct from those the counterfactual branch concentrates on. As demonstrated in Table 3, incorporating the loss function from Characteristic 3, our proposed LCA mechanism, namely CRNN\_FGNL (w/LCA), outperforms the previous state-of-the-art CRNN\_FGNL in all aspects, whether it be on original audio files or vocal-only tracks. However, we agree with your perspective that maximizing the differences in the attention maps between the counterfactual and main branches could inadvertently introduce inaccuracies. Therefore, our future research will focus on how to further discern the impact on classification from the regions that the counterfactual branch concentrates on.

---

> > ### Comment · Reviewer_hb43 · 2023-11-21
> > **Reply to the authors' rebuttal**
> >
> > Many thanks for your reply. I think your response partially solves my concerns. I decided to raise the score.

---

> > > ### Author Response · Authors · 2023-11-21
> > > **Response to the Reviewer hb43**
> > >
> > > Thank you very much for your positive feedback and for generously increasing the score.

---

### Meta-Review · Area_Chair_r9nG · 2023-12-08

**Metareview:**

This paper introduces a learnable counterfactual attention mechanism for singer identification, aiming to enhance the baseline method. While the authors present clear analyses and effective solutions, concerns arise about the logic behind certain design choices, particularly in Characteristics 2 and 3. Empirical evidence supporting assumptions and a more cautious approach to potential risks are warranted. The paper is well-structured, but addressing these concerns and providing a more rigorous experimental comparison with CAL would strengthen its contribution. The generality of the proposed method and the justification for focusing on the SID task need further clarification. Overall, improvements in experimental design and theoretical grounding are needed.

There are 3 valid reviewers with 2 rejections. After rebuttal, the 2 rejections were not yet convinced. AC read the concerns from the authors, but they are not strong justifications for the weakness of the paper. AC recommends reject.

**Justification For Why Not Higher Score:**

please see the weakness.

**Justification For Why Not Lower Score:**

N/A

---

### Decision · Program_Chairs · 2024-01-16

Reject